# Dose-response relationships of physical activity with bone mineral density and muscle mass in visceral obesity: A metabolic heterogeneity perspective

**Hongxin Zhuo, Xintao Zhang**⬭*

Department of Sports Medicine and Rehabilitation, Peking University Shenzhen Hospital, Shenzhen, Guangdong, China

* zhangxintao@sina.com

## Abstract

### Background

Obesity, particularly visceral adiposity, is associated with metabolic disorders and musculoskeletal deterioration. While physical activity (PA) benefits metabolic health, its dose-response effects on bone mineral density (BMD) and muscle mass in visceral obesity remain unclear.

### Objective

This study investigated associations between PA levels and musculoskeletal outcomes in visceral obesity, considering metabolic heterogeneity.

### Methods

Utilizing data from 3,077 visceral obesity participants (NHANES 2011–2018), PA levels were categorized as inactive (0 min/week), low- (1–150), moderate- (150–300), and high-active (>300). Weighted linear regression and restricted cubic splines analyzed associations, adjusted for demographic, socioeconomic, and health-related covariates. Subgroup analyses were conducted based on metabolic clusters, which were defined using k-means clustering according to blood pressure, visceral adiposity index, and HbA1c levels.

### Results

High-active PA correlated with increased lumbar BMD (β = 0.239, 95%CI:0.055–0.424, $P$ = 0.012), showing nonlinear saturation effects. Muscle mass improved dose-dependently with moderate-active (β: 0.021; 95%CI: 0.007–0.035; P = 0.005) and high-active (β:0.032; 95%CI: 0.018–0.046; P < 0.001). Subgroups with favorable

**Data availability statement:** All NHANES datasets used in the current analysis are available for download from the NHANES website: https://wwwn.cdc.gov/nchs/nhanes/Default.aspx.

**Funding:** This study was financially supported by the National Natural Science Foundation of China (No.82272568), the Guangdong Basic and Applied Basic Research Foundation (No.2022A1515220168), the Sanming Project of Medicine in Shenzhen (No.SZSM202211019) and the Shenzhen Science and Technology Program (No.KCXFZ20230731094100001), awarded to Zhang XT. The funders had no role in study design, data collection and analysis, decision to publish, or preparation of the manuscript.

**Competing interests:** The authors have declared that no competing interests exist.

**Abbreviations:** PA, physical activity; BMD, bone mineral density; NHANES, National Health and Nutrition Examination Survey; DXA, dual-energy X-ray absorptiometry; FNIH, the Foundation for the National Institutes of Health; ALM, appendicular lean mass; VAT, Visceral adipose tissue; SAT, subcutaneous adipose tissue; SBP, systolic blood pressure; DBP, diastolic blood pressure; TC, total cholesterol; VD, vitamin D levels; HbA1C, glycohemoglobin; VAI, visceral adiposity index; BMI, body mass index; RCS, restricted cubic splines; PIR, poverty-to-income ratio.

metabolic profiles exhibited stronger PA-musculoskeletal benefits, while metabolic dysfunction attenuated these associations.

## Conclusions

Bone adaptations require high-intensity PA thresholds, whereas muscle mass responds linearly to PA dose. Metabolic status modulates both relationships. Integrating metabolic optimization into exercise strategies is critical for improving musculoskeletal health in visceral obesity.

## 1 Introduction

Obesity, a multifaceted chronic disease, has escalated into a global epidemic with profound implications for public health [1]. The epidemiological landscape of obesity presents a stark reality: worldwide, nearly one-third of the world's population is now classified as overweight or obese. Projections indicate that by 2030, this figure could soar to over half of the world's population [2]. In recent years, emphasis has been placed on the heterotopic distribution of fat, and the abnormal accumulation of visceral fat is strongly associated with metabolic syndrome, type 2 diabetes mellitus and cardiovascular disease. Therefore, visceral obesity has become an important focus of obesity research [3–5].

There is an important connection between muscle, bone and fat, and this connection can be affected by age [6]. Muscle mass and bone mineral density (BMD) initially increase with age and then decline after maintaining a peak range for some time. When the degree of decline exceeds a certain threshold, sarcopenia and osteoporosis can occur [7,8].Studies in recent years have shown that obese people may co-exist with decreased muscle mass and/or BMD, constituting a complex condition of comorbidities [9–11]. The traditional view holds that obesity, defined by an elevated body mass index (BMI), exerts a greater mechanical load on the increased body weight to protect against the loss of BMD [12]. However, recent studies have found that although the BMD of obese patients is normal or above the normal level, their risk of fractures still increases, which is closely related to visceral fat [13].

It is worth noting that an increase in visceral fat, in particular, has been shown to be closely associated with a decrease in muscle mass [14] and BMD [15,16]. Visceral adipose tissue not only produces more inflammatory factors during metabolism, but also influences bone and muscle metabolism by secreting a variety of bioactive substances, such as leptin and adiponectin [17]. Moreover, the chronic inflammation mediated by these adipose tissues affects nutrition and lifestyle, which in turn alters BMD and muscle mass [18].

Numerous empirical studies have substantiated the beneficial impacts of PA and exercise on overall health outcomes. Proper and regular PA can help improve physiological function, prevent cardiovascular and metabolic disease risk, reduce obesity, and decrease BMD and muscle mass in elderly individuals [19]. Reducing sedentary time and appropriately increasing moderate to vigorous PA are associated with less

fat [20,21]. Recent studies have shown that regular PA can improve metabolic health in older adults, including lowering blood sugar, lipids, and blood pressure, while regulating gene expression and signaling pathways to reduce inflammation and oxidative stress [22,23]. These combined effects help slow the aging process and improve quality of life.

Although the improved effects of PA on metabolic health are well established at the scientific level, significant gaps remain at the level of public practice. Statistics show that less than a quarter of adults in the United States meet the recommended levels of PA, and this widespread sedentary lifestyle creates a vicious cycle of visceral fat accumulation [24,25]. Notably, existing studies have long relied on BMI as the core indicator when assessing obesity-related health risks, and this single-dimensional evaluation system has difficulty reflecting the special pathological mechanism of visceral fat ectopic deposition [3]. More critically, there is no clear dose–effect relationship between the PA level and improvements in the musculoskeletal system in the visceral obesity population, in particular, whether the metabolic level of the body affects the benefits of PA remains unknown. On the basis of these, this study aims to systematically explore the associations between PA levels and BMD and muscle mass in the visceral obesity population through multidimensional body component analysis to provide a basis for formulating precise exercise intervention strategies.

## 2 Methods

### 2.1 Study population

This study utilized personal data from the National Health and Nutrition Examination Survey (NHANES), a program conducted by the National Center for Health Statistics. The NHANES recruits a representative sample of the noninstitutionalized civilian population of the United States through a stratified, multistage random sampling approach. Since 1999, NHANES has been conducting a series of cross-sectional surveys biennially. Data collection encompasses both interviews and physical examinations. All participants were anonymized. For further details regarding the NHANES protocol, study design, and data collection methods, please refer to the official documentations.

The data analyzed in this study cover 2011–2012, 2013–2014, 2015–2016, and 2017–2018. Among 39156 NHANES participants from 2011 to 2018, rigorous assessments were conducted to ensure the completeness, consistency and logical coherence of PA data. In this study, participants were eliminated because they had inadequate information for determining the conditions of BMD and muscle mass (n = 21863). The dataset was then carefully examined to exclude individuals with missing questionnaire data on PA (n = 3523). In addition, participants lacking covariate data or who could not be classified as having visceral obesity (n = 10693) were excluded. Ultimately, 3077 participants had no missing or confounding information in terms of key outcomes, exposures, or variables (Fig 1).

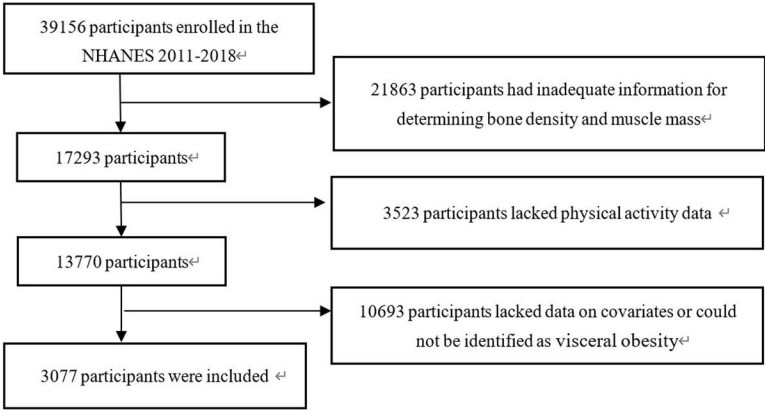

**Fig 1. Study participants flow chart.**

All participants provided written informed consent, and the data collection and research of NHANES were approved by the Ethics Review Board of the National Center for Health Statistics (Protocol Number: protocol #2005−06, protocol #2011−17 and protocol #2018−01).

## 2.2 Definition of visceral obesity

Visceral adipose tissue (VAT) and subcutaneous adipose tissue (SAT) were quantified via dual-energy X-ray absorptiometry (DXA), and the VAT/SAT ratio was subsequently calculated. Anthropometric measurements were conducted by certified technicians at mobile examination centers following standardized protocols. Visceral obesity was defined according to established diagnostic criteria: either a VAT/SAT ratio exceeding 1.0 or waist circumference measurements surpassing sex-specific thresholds (men: > 102 cm; women: > 88 cm) [10,26].

## 2.3 Definition of decreased BMD and muscle mass

The assessment of muscle mass was conducted in accordance with the standardized protocol established by the Foundation for the National Institutes of Health (FNIH) [27]. This methodology involves the calculation of the appendicular lean mass (ALM) to BMI ratio, which serves as a reliable indicator of muscle mass status. The ALM in this study was obtained by DXA. Specifically, muscle mass reduction is clinically defined when the ALM/BMI ratio falls below the sex-specific thresholds: < 0.789 for male subjects and <0.512 for female subjects.

For the evaluation of BMD, the study adopted a standardized measurement protocol from the NHANES guidelines [28]. The reference values were derived from the DXA Manufacturer's reference database, utilizing the mean and standard deviation of lumbar spine BMD measurements from a normative population of 30-year-old non-Hispanic white women. In accordance with the established diagnostic criteria, decreased BMD was identified when the T-score at the lumbar spine measurement site was ≤ −1.0 standard deviations below the reference mean [29].

## 2.4 Physical activity

Participants' PA levels were assessed using the Global Physical Activity Questionnaire, which inquired about the intensity, frequency, and duration of their PA over the preceding seven days. We screened for information on vigorous recreational activity and moderate recreational activity. Vigorous recreational activities are characterized by significant increases in breathing or heart rate, encompassing any intense exercise, fitness, or recreational pursuits. In contrast, moderate recreational activities result in only a modest elevation in breathing or heart rate. To standardize the measurement across different activity intensities, according to the WHO 2020 Physical Activity Guideline [30], the duration of vigorous activity is multiplied by 2 and then added to the duration of moderate activity to form the "equivalent duration".. Each participant's PA level was based on the "equivalent duration" of PA in minutes per week.. Referring to previous studies [31], participants were stratified into four distinct categories: inactive (0 minutes/week), low-active (1–150 minutes/week), moderate-active (150–300 minutes/week), and high-active (>300 minutes/week).

## 2.5 Covariate assessment

The study collected comprehensive demographic, socioeconomic, and health-related characteristics based on previous studies. Demographic variables included age (years), sex (female/male), and race/ethnicity (Mexican American, non-Hispanic White, non-Hispanic Black, other Hispanic, other race). Socioeconomic status was assessed through education level (high school or lower, college or higher), marital status (married/ living with partner, unmarried/ separated/ widowed), and poverty-to-income ratio (PIR; < 1.3, 1.3–3.5, or >3.5).

Health-related variables included both behavioral and clinical indicators. Behavioral factors included smoking status (defined as having smoked ≥100 cigarettes in lifetime) and alcohol consumption (classified according to sex-specific

criteria: ≤ 1 drink/day for women and ≤2 drinks/day for men) [32]. Sleep duration was categorized into three groups: < 7 hours, 7–9 hours, or >9 hours per night. Clinical measurements included systolic blood pressure (SBP) and diastolic blood pressure (DBP), total cholesterol (TC), vitamin D (VD), and glycohemoglobin (HbA1c) levels, as well as the diagnosis of related diseases such as depression, hypertension, and other comorbidities.

Depression is screened by the Patient Health Questionnaire-9 (PHQ-9) [33], with a cutoff score of ≥10 indicating clinically significant depression. Hypertension was defined as SBP ≥ 140 mmHg or DBP ≥ 90 mmHg. Information on smoking, drinking and sleep duration was also collected through standardized questionnaires administered by trained interviewers. The assessment criteria for comorbidities were modified based on the items of the Charles Comorbidities Index [34]. The final items include myocardial infarction, congestive heart failure, stroke, emphysema, chronic pulmonary bronchitis, arthritis, gout, liver disease, diabetes, gastrointestinal diseases, kidney diseases, and cancer or malignant tumors.

### 2.6  Statistics analysis

All data analyses take into account the complex sampling and weighting of NHANES data. Continuous variables are expressed as mean (standard deviation) and categorical variables are expressed as counts (percentage). Intergroup difference analysis of continuous variables was performed using one-way ANOVA. Chi-square tests or Fisher exact tests was used for categorical variables. To examine the associations between PA levels and both BMD and muscle mass, we employed weighted linear regression models with a hierarchical adjustment approach. Three sequential models were constructed: Model1 represented the unadjusted crude analysis; Model2 incorporated adjustments for age and sex; and Model3 further adjusted for race, education, sleep duration, depression, hypertension, TC, VD, smoking, PIR and comorbidities in addition to the covariates in Model2. The nonlinear relationship between PA and BMD and muscle mass was studied with weighted restricted cubic splines (RCS).

To enhance the robustness of our findings and investigate potential effect modifications, we conducted two sensitivity analyses. First, we performed a quartile-based stratification of PA levels among highly active individuals to examine whether extremely high levels of PA conferred additional benefits. Second, to investigate potential effect modification by metabolic health status, we performed k-means cluster analysis using the key metabolic parameters: SBP, DBP, visceral adiposity index (VAI) [35], and HbA1c. This unsupervised learning method grouped participants based on similarity across these standardized variables, aiming to identify distinct metabolic phenotypes within the visceral obesity population. Subsequent subgroup analyses were conducted to determine whether the effects of PA on muscle mass and BMD were modified by metabolic status.

## 3  Results

### 3.1  Baseline characteristics

The study included a total of 3,077 participants, with a mean age of 40.54 ± 10.98 years (Table 1). The cohort consisted of 1,844 females (59.9%) and 1,233 males (40.1%). Regarding lifestyle factors, 45.0% of participants reported smoking, whereas 60.2% reported alcohol consumption. A notable age-related trend was observed, with younger individuals demonstrating significantly higher levels of PA engagement. The prevalence rates of depression and hypertension within the study population were 9.8% and 37.3%, respectively. There were significant differences in age, sex, race, education, depression, hypertension, PIR, sleep duration, smoking, VD, TC and comorbidities among people with different levels of PA (P < 0.05).

### 3.2  Associations of BMD and PA

In Model 1,2, both low-active and high-active PA levels were positively correlated with lumbar BMD compared to the inactive level (Table 2). However, after comprehensive adjustment for multiple covariates in Model 3, only the high

**Table 1. Baseline characteristics.**

| | Overall | Inactive | Low-active | Moderate-active | High-active | P |
|---|---|---|---|---|---|---|
| Counts | 3077 | 1474 | 508 | 372 | 723 | |
| Age (years) | 40.54 (10.98) | 41.74 (10.78) | 41.14 (10.52) | 40.25 (11.00) | 37.84 (11.24) | <0.001 |
| Sex (%) | | | | | | 0.008 |
| Female | 1844 (59.9) | 876 (59.4) | 332 (65.4) | 231 (62.1) | 405 (56.0) | |
| Male | 1233 (40.1) | 598 (40.6) | 176 (34.6) | 141 (37.9) | 318 (44.0) | |
| Race (%) | | | | | | <0.001 |
| Mexican American | 560 (18.2) | 314 (21.3) | 65 (12.8) | 64 (17.2) | 117 (16.2) | |
| Non-Hispanic Black | 715 (23.2) | 338 (22.9) | 114 (22.4) | 91 (24.5) | 172 (23.8) | |
| Non-Hispanic White | 1220 (39.6) | 547 (37.1) | 223 (43.9) | 164 (44.1) | 286 (39.6) | |
| Other Hispanic | 304 (9.9) | 163 (11.1) | 48 (9.4) | 23 (6.2) | 70 (9.7) | |
| Other Race | 278 (9.0) | 112 (7.6) | 58 (11.4) | 30 (8.1) | 78 (10.8) | |
| Education (%) | | | | | | <0.001 |
| College+ | 1911 (62.1) | 758 (51.4) | 366 (72.0) | 275 (73.9) | 512 (70.8) | |
| High school- | 1166 (37.9) | 716 (48.6) | 142 (28.0) | 97 (26.1) | 211 (29.2) | |
| Marital (%) | | | | | | 0.139 |
| Married | 1858 (60.4) | 867 (58.8) | 315 (62.0) | 242 (65.1) | 434 (60.0) | |
| Unmarried | 1219 (39.6) | 607 (41.2) | 193 (38.0) | 130 (34.9) | 289 (40.0) | |
| Depression (%) | | | | | | <0.001 |
| No | 2775 (90.2) | 1270 (86.2) | 465 (91.5) | 355 (95.4) | 685 (94.7) | |
| Yes | 302 (9.8) | 204 (13.8) | 43 (8.5) | 17 (4.6) | 38 (5.3) | |
| Hypertension (%) | | | | | | <0.001 |
| No | 1930 (62.7) | 879 (59.6) | 301 (59.3) | 240 (64.5) | 510 (70.5) | |
| Yes | 1147 (37.3) | 595 (40.4) | 207 (40.7) | 132 (35.5) | 213 (29.5) | |
| PIR (%) | | | | | | <0.001 |
| <1.3 | 949 (30.8) | 579 (39.3) | 115 (22.6) | 76 (20.4) | 179 (24.8) | |
| >3.5 | 941 (30.6) | 320 (21.7) | 198 (39.0) | 149 (40.1) | 274 (37.9) | |
| 1.3-3.5 | 1187 (38.6) | 575 (39.0) | 195 (38.4) | 147 (39.5) | 270 (37.3) | |
| Sleep duration (%) | | | | | | 0.018 |
| <7 | 1117 (36.3) | 546 (37.0) | 182 (35.8) | 120 (32.3) | 269 (37.2) | |
| >9 | 133 (4.3) | 82 (5.6) | 17 (3.3) | 11 (3.0) | 23 (3.2) | |
| 7-9 | 1827 (59.4) | 846 (57.4) | 309 (60.8) | 241 (64.8) | 431 (59.6) | |
| Smoking (%) | | | | | | <0.001 |
| No | 1693 (55.0) | 730 (49.5) | 284 (55.9) | 224 (60.2) | 455 (62.9) | |
| Yes | 1384 (45.0) | 744 (50.5) | 224 (44.1) | 148 (39.8) | 268 (37.1) | |
| Drinking (%) | | | | | | 0.277 |
| No | 1224 (39.8) | 566 (38.4) | 220 (43.3) | 150 (40.3) | 288 (39.8) | |
| Yes | 1853 (60.2) | 908 (61.6) | 288 (56.7) | 222 (59.7) | 435 (60.2) | |
| VD (nmol/L) | 54.72 (24.23) | 51.89 (22.89) | 56.29 (24.70) | 57.09 (24.86) | 58.17 (25.59) | <0.001 |
| TC (mg/dL) | 195.67 (41.18) | 196.54 (41.55) | 198.53 (44.40) | 197.22 (40.02) | 191.10 (38.31) | 0.006 |
| Comorbidities | 0.80 (1.16) | 0.92 (1.22) | 0.81 (1.16) | 0.70 (1.08) | 0.62 (1.03) | <0.001 |

Mean±SD for continuous variables; percentage for categorical variables; P-value was calculated by the weighted data; VD: vitamin D; TC: total cholesterol; PIR, poverty-to-income ratio.

**Table 2. Relationship between BMD, muscle mass and PA in weighted multiple linear regression.**

| | Model 1 β (95% CI) P value | Model 2 β (95% CI) P value | Model 3 β (95% CI) P value |
|---|---|---|---|
| **BMD** | | | |
| Inactive | Ref | Ref | Ref |
| Low-active | 0.164 (0.024-0.303) 0.022 | 0.144 (0.002-0.286) 0.046 | 0.115 (−0.031-0.261) 0.120 |
| Moderate-active | 0.041 (−0.174-0.257) 0.702 | 0.021 (−0.192-0.233) 0.846 | −0.019 (−0.231-0.192) 0.854 |
| High-active | 0.315 (0.12-0.51) 0.002 | 0.304 (0.107-0.501) 0.003 | 0.239 (0.055-0.424) 0.012 |
| | | | |
| **Muscle mass** | | | |
| Inactive | Ref | Ref | Ref |
| Low-active | −0.005 (−0.031-0.02) 0.675 | 0.017 (0.004-0.031) 0.015 | 0.005 (−0.008-0.018) 0.476 |
| Moderate-active | 0.018 (−0.011-0.046) 0.221 | 0.036 (0.022-0.05) <0.001 | 0.021 (0.007-0.035) 0.005 |
| High-active | 0.05 (0.026-0.074) <0.001 | 0.048 (0.034-0.062) <0.001 | 0.032 (0.018-0.046) <0.001 |

Model1: unadjusted;

Model2: adjusted age and sex;

Model3: adjusted age, sex, race, education, comorbidities, sleep duration, depression, hypertension, TC, VD, smoking and PIR.

PA level maintained a statistically significant positive association with lumbar BMD (β: 0.239; 95%CI: 0.055–0.424; P = 0.012 < 0.05). To further investigate the dose-response relationship, we conducted a stratified analysis of high-active PA levels by dividing them into quartiles based on PA duration: Q1 (300–360 min), Q2 (360–540 min), Q3 (540–790 min), and Q4 (>790 min) (Table 3). No statistically significant differences were observed across these quartiles (P > 0.05). As shown in Fig 2, there is a significant nonlinear relationship between lumbar BMD and PA (P for overall < 0.001, P for non-linear <0.001), suggesting a complex dose-response pattern between PA and BMD outcomes.

## 3.3 Associations of muscle mass and PA

The association between PA levels and muscle mass was systematically evaluated across three statistical models (Table 2). In Model 1, the analysis revealed that only high-active PA levels demonstrated a significant positive association with muscle mass. In Model 2, all PA levels are positively correlated. Finally, analysis in Model 3 provided refined estimates. Moderate-active (β: 0.021; 95%CI: 0.007–0.035; P = 0.005 < 0.05) and high-active (β:0.032; 95%CI: 0.018–0.046; P < 0.001) PA levels

**Table 3. Comparative analysis of advanced PA Levels.**

| | Model 1 β (95% CI) P value | Model 2 β (95% CI) P value | Model 3 β (95% CI) P value |
|---|---|---|---|
| **BMD** | | | |
| Q1 | Ref | Ref | Ref |
| Q2 | 0.218 (−0.170-0.607) 0.264 | 0.213 (−0.179-0.604) 0.282 | 0.209 (−0.140-0.557) 0.233 |
| Q3 | 0.152 (−0.225-0.529) 0.422 | 0.142 (−0.230-0.513) 0.448 | 0.168 (−0.201-0.537) 0.364 |
| Q4 | 0.080 (−0.244-0.403) 0.625 | 0.060 (−0.262-0.382) 0.709 | 0.108 (−0.209-0.425) 0.494 |
| **Muscle mass** | | | |
| Q1 | Ref | Ref | Ref |
| Q2 | 0.031 (−0.012-0.075) 0.15 | 0.024 (−0.002-0.049) 0.071 | 0.026 (0.002-0.050) 0.033 |
| Q3 | 0.036 (−0.013-0.085) 0.145 | 0.034 (0.008-0.06) 0.011 | 0.028 (0.004-0.052) 0.024 |
| Q4 | 0.067 (0.021-0.114) 0.005 | 0.032 (0.004-0.06) 0.026 | 0.036 (0.012-0.060) 0.005 |

Model1: unadjusted;

Model2: adjusted age and sex;

Model3: adjusted age, sex, race, education, comorbidities, sleep duration, depression, hypertension, TC, VD, smoking and PIR.

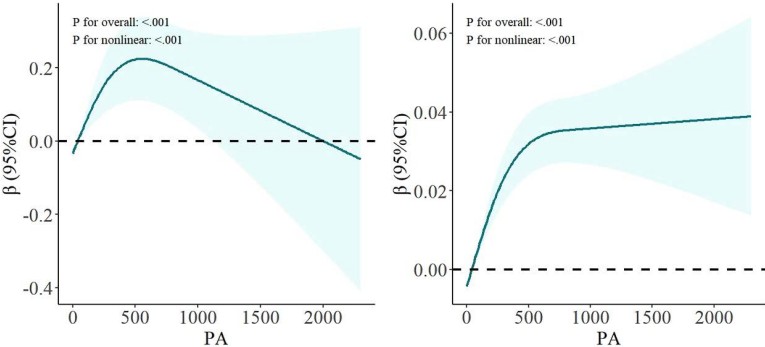

**Fig 2. Non-linear relationship of PA and BMD (left), muscle mass (right).** Models were adjusted for age, sex, race, education, comorbidities, sleep duration, depression, hypertension, TC, VD, smoking and PIR.

were positively correlated with muscle mass. Further stratification analysis revealed statistically significant differences across all interval levels when compared to the Q1 (Table 3). As shown in Fig 2, a significant nonlinear relationship was observed between muscle mass and PA levels (P for overall<0.001, P for nonlinear<0.001).

### 3.4 Subgroup analysis

In the subgroup analysis, people with visceral obesity identified using body measurement data were further subdivided. Subgroups 1–4 were obtained by k-means based on SBP, DBP, VAI, and HBA1c. Table 4 presents the comparative characteristics of these four indicators across the subgroups, demonstrating statistically significant intergroup differences (p<0.05). Subgroup analysis was referenced to model 3 and included remaining covariates other than hypertension and TC. The results revealed differential associations between PA levels and BMD and muscle mass across subgroups (Fig 3). Regarding BMD, a significant positive association was observed exclusively in subgroup 1 among individuals engaging in high-active PA. For muscle mass, positive correlations were identified in subgroups 1 and 2 for participants maintaining moderate-active and high-active PA levels. Conversely, no significant associations were detected between PA and either BMD or muscle mass in Subgroups 3 and 4 (p>0.05).

## 4 Discussion

This study elucidates the complex relationships between PA, BMD, and muscle mass, while highlighting the moderating role of metabolic health in these associations within the context of visceral obesity. Previous studies have found that PA is positively correlated with lumbar spine BMD and muscle mass [36,37]. However, this study reveals that when the research scope is confined to individuals with visceral obesity, a notable threshold effect becomes evident: a weekly PA duration exceeding 150 minutes exhibits a protective effect on muscle mass, while a duration surpassing 300 minutes

**Table 4. Subgroup characteristics.**

| Subgroup | 1 | 2 | 3 | 4 | P |
|---|---|---|---|---|---|
| SBP | 106.68 (7.24) | 118.06 (5.77) | 131.97 (6.37) | 156.84 (13.06) | <0.001 |
| DBP | 61.75 (9.05) | 72.55 (6.29) | 81.00 (7.48) | 91.1 (12.03) | 0.002 |
| VAI | 5.47 (4.89) | 7.13 (8.55) | 7.76 (8.97) | 7.98 (8.14) | <0.001 |
| HbA1c | 5.56 (0.98) | 5.68 (1.06) | 5.92 (1.28) | 6.19 (1.42) | <0.001 |

Each variable is expressed as the Mean (SD).

Differences between groups were obtained by ANOVA.

| Subgroup | PA | β (95% CI) | P value | Subgroup | PA | β (95% CI) | P value |
|---|---|---|---|---|---|---|---|
| 1 | Inactive | 0 | | 1 | Inactive | 0 | |
| | Low-active | 0.2 (-0.195-0.594) | 0.313 | | Low-active | 0.033 (0.002-0.064) | 0.036 |
| | Moderate- active | 0.117 (-0.299-0.533) | 0.573 | | Moderate- active | 0.044 (0.015-0.073) | 0.004 |
| | High-active | 0.507 (0.154-0.859) | 0.006 | | High-active | 0.06 (0.037-0.083) | <0.001 |
| 2 | Inactive | 0 | | 2 | Inactive | 0 | |
| | Low-active | 0.039 (-0.173-0.251) | 0.713 | | Low-active | -0.003 (-0.023-0.017) | 0.771 |
| | Moderate- active | -0.155 (-0.428-0.118) | 0.259 | | Moderate- active | 0.019 (-0.001-0.038) | 0.06 |
| | High-active | 0.086 (-0.154-0.327) | 0.473 | | High-active | 0.028 (0.007-0.049) | 0.012 |
| 3 | Inactive | 0 | | 3 | Inactive | 0 | |
| | Low-active | 0.148 (-0.128-0.423) | 0.286 | | Low-active | -0.009 (-0.033-0.015) | 0.444 |
| | Moderate- active | 0.156 (-0.191-0.503) | 0.37 | | Moderate- active | 0.014 (-0.01-0.038) | 0.249 |
| | High-active | 0.29 (-0.158-0.739) | 0.199 | | High-active | 0.013 (-0.011-0.036) | 0.29 |
| 4 | Inactive | 0 | | 4 | Inactive | 0 | |
| | Low-active | 0.075 (-0.621-0.77) | 0.822 | | Low-active | -0.001 (-0.054-0.052) | 0.96 |
| | Moderate- active | -0.371 (-1.272-0.529) | 0.393 | | Moderate- active | 0.009 (-0.052-0.07) | 0.766 |
| | High-active | -0.223 (-0.838-0.392) | 0.452 | | High-active | 0.041 (-0.021-0.103) | 0.184 |

**Fig 3. Subgroup analyses of the association between PA and BMD (left), muscle mass (right).** Models were adjusted for age, sex, race, education, comorbidities, sleep duration, depression, VD, smoking and PIR.

demonstrates a protective effect on lumbar spine BMD. Further subgroup analysis of metabolic levels revealed that there were significant differences in the effects of PA on BMD and muscle mass under different metabolic states. The beneficial effects of PA on BMD and muscle mass were only found in subgroups with better metabolic status.

The findings demonstrate that excessive accumulation of VAT may mask or counteract the synthetic effect of PA on the musculoskeletal system. This may be related to the interference of chronic inflammatory status and insulin resistance in the context of metabolic abnormalities. First, chronic low-grade inflammation may weaken the anabolic effect of mechanical stimulation, and pro-inflammatory factors such as IL-6 and TNF-α secreted by visceral adipose tissue can inhibit IGF-1 signaling pathway and interfere with exercise-induced muscle protein synthesis and bone formation [38]. In addition, insulin resistance prevalent in people with high VAI may impair exercise-induced IGF-1 signaling and affect skeletal muscle hypertrophy and bone matrix mineralization processes [39,40]. Importantly, lipid toxicity caused by visceral fat accumulation may alter mesenchymal stem cell differentiation via the PPARγ pathway to favor adipocyte differentiation over osteoblast differentiation [41], and this shift in cell fate determination may partially explain the diminished benefits of PA in the metabolic disorders subgroup.

Furthermore, the results of the dose-response relationship of PA demonstrated the heterogeneity of PA in BMD and muscle mass. The incremental gain of BMD decreases after the optimal PA duration. Excessive increase in PA time does not bring additional benefits and may even counteract the advantages brought by PA, forming an inverted "U "curve. This is similar to the finding of Huang et al. in physical exercise among sedentary elderly people, where high-dose exercise can have negative effects [42]. Unlike the case of BMD, there is no evidence suggesting that higher doses of PA have a counteracting effect in this research. This indicates that for muscle mass, the positive correlation with PA persists at different activity levels. The heterogeneity of this benefit reflects the key differences in the underlying biology of muscle tissue and bone.

Mechanical stimulation is transduced into electrochemical signals via mechanosensitive channels, such as piezoelectric channels. This transduction subsequently activates key signaling pathways, including the Wnt/β-catenin pathway [43]. Moreover, under the mediation of G protein-coupled receptors and integrins, the mechanotransduction process further modulates load-induced bone metabolism, thereby facilitating bone formation. But excessive exercise can overload the bones, causing the rate of bone resorption to exceed the rate of bone formation and resulting in a decrease in BMD [44].

Muscle cells promote protein synthesis by activating the mTOR signaling pathway under mechanical stress, and this process is further amplified by growth factors such as IGF-1 [45]. Moreover, the activation mechanism of skeletal muscle satellite cells shows the characteristic of continuous response under continuous loading, which may be an important basis for the continuous improvement of muscle mass with the increase of exercise dose [46,47].

Body composition measurement has become the core of research in nutrition, exercise and metabolic health [48]. PA plays a pivotal role in safeguarding individual health by regulating the proportion of body composition [49]. Notably, the accumulation of VAT may have an impact on this regulatory mechanism by altering the metabolic state [17]. Therefore, in order to go beyond the basic understanding of body composition as a health indicator, it is necessary to transform these complex dose-response relationships and metabolic heterogeneity into targeted intervention strategies. This study suggests that for exercise prescriptions for people with visceral obesity, the dose of PA must reach the corresponding threshold. Meanwhile, merely changing the amount of PA may not be sufficient to improve the performance of the muscu-loskeletal system. Priority should be given to controlling the metabolic state.

The main advantages of this study are reflected in several levels. Firstly, the study design is based on the nationally representative NHANES database and adopts stratified multi-stage random sampling method to ensure the broad rep-resentation of the samples and the universal applicability of the results. The uniqueness of the research perspective is reflected in the simultaneous attention to the dual outcome indicators of BMD and muscle mass in visceral obesity, and the systematic exploration of the regulatory effects of different metabolic states on exercise effects, which provides new insights into the heterogeneous effects of exercise intervention in the context of metabolic syndrome. In addition, by introducing cluster analysis of visceral fat index and metabolic parameters, the study achieved a multidimensional char-acterization of metabolic characteristics of visceral obesity population, which strengthened the biological rationality of the results of subgroup analysis.

Despite these insights, several limitations warrant consideration. PA assessments rely on self-reported questionnaires, which may be subject to recall bias and social expectation bias, especially regarding the accuracy of the duration and frequency of high-intensity activities. And due to the applicability of the questionnaire items, we are unable to delve the suggestions into the types of exercise. Furthermore, it is important to note that the study population, constrained by the NHANES measurement standards for BMD and muscle mass, predominantly comprises young and middle-aged individ-uals, thereby limiting the generalizability of findings to the elderly population. Moreover, the muscle mass assessment method of ALM/BMI cannot distinguish the distribution of fat in the body, and its application in the population with visceral obesity may have limitations. Finally, our metabolic clusters prioritized parameters with high data completeness (BP, VAI, HbA1c). Inflammatory markers (like CRP) and dynamic metabolic measures (like postprandial glucose) were excluded due to a high degree of data deficiency, potentially limiting cluster comprehensiveness. Cohort studies with protocolized biomarker collection are needed to validate these phenotypes.

Future research should prioritize disentangling the effects of specific exercise modalities on bone and muscle within population with visceral obesity, as their mechano-adaptive pathways differ. Mechanistic studies using multi-omics approaches could identify molecular networks disrupted by metabolic dysfunction, offering targets for adjuvant therapies to restore exercise responsiveness. Clinically, integrating PA with anti-inflammatory or insulin-sensitizing interventions may yield synergistic benefits in metabolically compromised individuals.

## 5 Conclusion

In conclusion, this study delineates distinct dose-response patterns of PA on BMD and muscle mass in the population with visceral obesity, which are regulated by potential metabolic health. These findings advocate for metabolic stratification in exercise prescriptions and underscore the importance of personalized PA prescriptions to optimize musculoskeletal out-comes across diverse populations. Addressing metabolic dysfunction may unlock PA's full potential as a therapeutic tool, bridging gaps between preventive health and precision medicine.

## Acknowledgments

The authors thank the NHANES team for their efforts and the cooperation of every participant.

## Author contributions

**Conceptualization:** Xintao Zhang.

**Formal analysis:** Hongxin Zhuo.

**Methodology:** Hongxin Zhuo, Xintao Zhang.

**Writing – original draft:** Hongxin Zhuo, Xintao Zhang.

**Writing – review & editing:** Hongxin Zhuo, Xintao Zhang.

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
