## [Decision Letter · Decision Letter 0]

PONE-D-25-12300Dose-response relationships of physical activity with bone mineral density and muscle mass in visceral obesity: A metabolic heterogeneity perspectivePLOS ONE

Dear Dr. zhang,

Thank you for submitting your manuscript to PLOS ONE. After careful consideration, we feel that it has merit but does not fully meet PLOS ONE’s publication criteria as it currently stands. Therefore, we invite you to submit a revised version of the manuscript that addresses the points raised during the review process.

We look forward to receiving your revised manuscript.

Kind regards,

Toshio Matsumoto

Academic Editor

PLOS ONE

Journal Requirements:

“This study was supported by grant from the Shenzhen Science and Technology Program (KCXFZ20230731094100001)”

Additional Editor Comments:

As indicated in the comments by the three expert reviewers, although the manuscript contains some useful information, there are some issues that need to be addressed before the manuscript can be considered for publication in PLOS ONE. These include, but are not limited to, the below:

1. Physical activity levels are categorized into four levels from inactive to high-active based upon only time length per a week. However, the type and intensity of activity should be taken into account. It is important to propose which type or intensity of exercise is recommended to improve BMD and/or muscle mass.

2. Subgroup analyses were performed based upon SBP, DBP, VAI, and HbA1c. What is the rationale for using these four parameters? Is there a possibility that more comprehensive analyses including other metabolic can give better results?

3. Muscle mass is not directly measured but is estimated from the ratio of the appendicular lean mass to body mass index. This should be mentioned in the limitation.

4. Exclusion criteria is not mentioned. It is unclear whether the patients were treated for hypertension, diabetes, obesity, etc.

Reviewers' comments:

Reviewer's Responses to Questions

**Comments to the Author**

1. Is the manuscript technically sound, and do the data support the conclusions?

Reviewer #1: Partly

Reviewer #2: Partly

Reviewer #3: Yes

2. Has the statistical analysis been performed appropriately and rigorously? 

Reviewer #1: I Don't Know

Reviewer #2: I Don't Know

Reviewer #3: Yes

3. Have the authors made all data underlying the findings in their manuscript fully available?

Reviewer #1: Yes

Reviewer #2: Yes

Reviewer #3: Yes

4. Is the manuscript presented in an intelligible fashion and written in standard English?

Reviewer #1: Yes

Reviewer #2: Yes

Reviewer #3: Yes

5. Review Comments to the Author

Reviewer #1: 1. The reference number ‘12’ is not complete. Please correct it.

2. “Loss of bone and muscle mass often indicates accumulation of adipose tissuee[9]. Studies in recent years have shown that obese people may co-exist with decreased muscle mass and/or BMD, constituting a complex condition of comorbidities” The relation between obesity and osteoporosis is still debatable. Historically, obesity is protective against osteoporosis. Since this is still an ongoing thesis, further studies are needed, I would revise your statement.

3. I could not see a clear exclusion criteria. I am wondering if the patients with a history of cancer, or severe comorbidities were included into the study.

4. So, the authors define VAT and SAT and the importance of their ratios. I am wondering in which statistical analysis they used this definition. They used VAI for analysis, but it is totally a different measurement, right?

5. Because NHANES is cross-sectional, causality cannot be inferred (e.g., healthier individuals may exercise more). I am wondering what kind of additional data was used for analysis. Any comorbidity scoring index etc.

6. Clustering was performed on only 4 variables (SBP, DBP, VAI, HbA1c). I am wondering why only these 4 variables were chosen and whether cluster robustness was tested.

7. It is unclear if vitamin D levels, depression scores, and smoking were included in all models. Table 2 mentions these, but the Methods section should explicitly say which covariates are included in which models.

Reviewer #2: In this study, they investigated the relationships between physical activity category levels and bone mineral density at the lumbar spine and an appendicular muscle mass parameter in 3077 subjects with visceral obesity based on the NHANES data base.

Although the paper includes some useful information, there are several issues, which should be addressed.

1. The detailed and substantial contents of classification about physical activity levels should be described in Methods, although one reference [30] was only cited.

2. The content of Subgroup analysis seemed ambiguous, which details should be explained clearly.

3. In this study, the standard methods have not used for the assessment of bone mineral density and muscle mass. This limitation should be described in Discussion.

4. The results of this study should be discussed in association with visceral obesity.

Reviewer #3: General comments

The author reported the dose-response relationship of physical activity (PA) with bone mineral density (BMD) and muscle mass in visceral obesity. This relationship was modulated by metabolic status. The most important message of this study is that, for the patients with visceral obesity, it is crucial to take an individualized approach that considers their metabolic status in addition to the amount of exercise, in order to maintain healthy bones and muscles. This report has informative and significant contents at the actual clinical setting. There are several issues that the authors should adequately address in this manuscript.

Specific comments

1) PA levels were categorized as inactive (0 min/week), low-active (1-150), moderate-active (150-300), and high-active (>300) in this study. When examining the relationship between PA and BMD/muscle mass, it is important to consider the type of exercise such as aerobic exercise, resistance exercise, or combined exercise, as well as its intensity and duration. The details are unclear in surveys of PA based on self-reported questionnaires of this cross-sectional study. The issue with this paper is that, given the wide variety of available exercise options, it lacks specific recommendations that can be suggested to the patients.

2) Subgroup analyses were conducted based on metabolic clusters, which were defined using k-means clustering according to blood pressure, visceral adiposity index, and HbA1c levels. Although metabolic clusters are being classified using blood pressure, visceral fat index, and HbA1c levels, is this classification valid? How do the authors think about the opinion that a classification based on more comprehensive data including lipid profiles and inflammatory markers yield more accurate and valid results?

3) Is whether or not patients are receiving treatment for hypertension, dyslipidemia, or diabetes mellitus taken into consideration in this analysis?

6. PLOS authors have the option to publish the peer review history of their article (what does this mean? ). If published, this will include your full peer review and any attached files.

**Do you want your identity to be public for this peer review?** For information about this choice, including consent withdrawal, please see our Privacy Policy .

Reviewer #1: No

Reviewer #2: No

Reviewer #3: No

While revising your submission, please upload your figure files tohe manuscript record, and check for the action link "View Attachments". If this link does not appear, there are the Preflight Analysis and Conversion Engine (PACE) digital diagnostic tool, https://pacev2.apexcovantage.com/ . PACE helps ensure that figures meet PLOS requirements. To use PACE, you must first register as a user. Registration is free. Then, login and navigate to the UPLOAD tab, where you will find detailed instructions on how to use the tool. If you encounter any issues or have any questions when using PACE, please email PLOS at figures@plos.org . Please note that Supporting Information files do not need this step.

---

## [Author Response · Author response to Decision Letter 1]

13 Jun 2025

All the contents have been presented in the file "Response to Reviewers".

---

## [Decision Letter · Decision Letter 1]

Dose-response relationships of physical activity with bone mineral density and muscle mass in visceral obesity: A metabolic heterogeneity perspective

PONE-D-25-12300R1

Dear Dr. zhang,

We’re pleased to inform you that your manuscript has been judged scientifically suitable for publication and will be formally accepted for publication once it meets all outstanding technical requirements.

Kind regards,

Toshio Matsumoto

Academic Editor

PLOS ONE

Reviewers' comments:

Reviewer's Responses to Questions

**Comments to the Author**

1. If the authors have adequately addressed your comments raised in a previous round of review and you feel that this manuscript is now acceptable for publication, you may indicate that here to bypass the “Comments to the Author” section, enter your conflict of interest statement in the “Confidential to Editor” section, and submit your "Accept" recommendation.

Reviewer #2: (No Response)

Reviewer #3: All comments have been addressed

2. Is the manuscript technically sound, and do the data support the conclusions?

Reviewer #2: (No Response)

Reviewer #3: Yes

3. Has the statistical analysis been performed appropriately and rigorously? 

Reviewer #2: (No Response)

Reviewer #3: Yes

4. Have the authors made all data underlying the findings in their manuscript fully available?

Reviewer #2: (No Response)

Reviewer #3: Yes

5. Is the manuscript presented in an intelligible fashion and written in standard English?

Reviewer #2: (No Response)

Reviewer #3: Yes

6. Review Comments to the Author

Reviewer #2: (No Response)

Reviewer #3: (No Response)

7. PLOS authors have the option to publish the peer review history of their article (what does this mean? ). If published, this will include your full peer review and any attached files.

**Do you want your identity to be public for this peer review?** For information about this choice, including consent withdrawal, please see our Privacy Policy .

Reviewer #2: No

Reviewer #3: No

---

## [Editor Report · Acceptance letter]

PONE-D-25-12300R1

PLOS ONE

Dear Dr. Zhang,

I'm pleased to inform you that your manuscript has been deemed suitable for publication in PLOS ONE. Congratulations! Your manuscript is now being handed over to our production team.

Kind regards,

on behalf of

Dr. Toshio Matsumoto

Academic Editor

PLOS ONE